# Parks and Recreational Areas as Sinks of Plastic Debris in Urban Sites: The Case of Light-Density Microplastics in the City of Amsterdam, The Netherlands

Quirine M. Cohen [1], Mae Glaese [1,2], Ke Meng [1], Violette Geissen [1] and Esperanza Huerta-Lwanga [1,3,*]

1 Soil Physics and Land Management Degradation Group, Wageningen University & Research, 6708 PB Wageningen, The Netherlands; cohenquirine@gmail.com (Q.M.C.); l.glaese@ucr.nl (M.G.); ke.meng@wur.nl (K.M.); violette.geissen@wur.nl (V.G.)
2 Department of Environmental and Earth Science, University College Roosevelt, 4331 CB Middelburg, The Netherlands
3 Departamento de Agricultura, Sociedad y Ambiente, El Colegio de la Frontera Sur (ECOSUR), Campeche 24500, Mexico
* Correspondence: esperanza.huertalwanga@wur.nl

**Abstract:** Soils of parks and recreational areas are potential sinks of microplastics because they are under multifunctional use. The aims of this research were to quantify and determine the types and abundance of light-density microplastics in one of the most cosmopolitan cities of the world: Amsterdam, The Netherlands. Therefore, potential differences between the city districts were explored through the assessment of light-density microplastics' concentrations in soils together with the soil properties. Microplastics were extracted from 74 soil samples. Predictions of microplastic concentrations and soil characteristics were made for the entire city by using ordinary kriging; 97% of the samples contained microplastic particles (MPPs), and on average, there were 4825.31 ± 6513.85 MPP/kg soil. A total of 21 hotspot samples were identified, and all of them contained LDPE, which represented 40.82% of the plastic types, in addition to 35.06% PAC and 15.58% natural polyamide. Other types of plastics were PP (0.19%), PS (1.30%), bioplastic (0.19%), PA (0.37%), PU (0.56), PVC (0.19%), and unidentified plastics (0.19%). There were no significant differences in MPP concentration between city districts. Our results showed that MPPs are abundant in urban soils, which represents a high risk for soil life. Further studies are required for identifying the sources of this pollution.

**Keywords:** microplastics; urban areas; parks

## 1. Introduction

It is predicted that the current worldwide production of plastic will double in the coming 20 years, extending the post-consumer waste [1]. When not collected and processed correctly, it is evident that plastic debris will end up in the environment [2]. Microplastic pollution has been researched extensively in the marine environment [3] and to some degree in the terrestrial environment [4–6]. However, research on microplastic pollution in an urban environment has been lacking [7]. This is remarkable, since cities can contain plastic debris in their soil itself, and they be a source for plastic debris outside the urban areas [6,8].

Since most plastic litter comes from land, it is highly likely that these plastics have first interacted with the terrestrial ecosystem [9]. Microplastics are small particles of <5 mm that are insoluble in water, non-degradable, and have chemical properties that influence their availability to organisms [10]. Studies have shown that microplastic pollution is related to its degradation rate, which is generally slow or does not even occur [4,11]. These particles can be transported through the soil both horizontally and vertically through anthropogenic activities or naturally by either leaching or earthworms [4,12]. Moreover, microplastics

can increase mortality rates and decrease reproduction rates, in addition to reducing the growth of earthworms [13].

There are multiple ways that plastic debris can end up in an urban environment. Agricultural practices are suggested to be the main contributor and could be a source of plastic debris in urban soils through atmospheric transportation [6,14]. Other sources include release from (1) transportation, which can occur from damaged vehicles or road channelizing devices [7], (2) littering by individuals after using plastic materials [15], (3) deposition after plastic products [16] or synthetic fibres from clothes or houses leach into the atmospheric compartment [17], and, finally, (4) plastic debris entering the terrestrial environment after tourism-related land use. Amsterdam is famous for its eight canals within the city centre, which are part of the UNESCO World Heritage [18], making it a popular tourist destination. Moreover, tourism is increasing each year due to Amsterdam's open-minded reputation and marketing efforts from the city itself, which leads to more littering [19]. Another tourism-related land use is from events, such as festivals. Festivals are typically large public events that last for a short time frame [20]. In The Netherlands, 50 million kg of plastic garbage is produced by all festivals each year [21]. Since Amsterdam hosts most of the festivals in The Netherlands (192 in 2018) [21], this could be a likely source.

Currently, 862,987 people are living in Amsterdam, the capital of The Netherlands [22]. In 2018, 3.5 kg plastic garbage was produced per person [23]. Amsterdam lies next to the North Sea and is surrounded by nature reserves that are part of the UNESCO World Heritage [24]. Due to its (culturally valued) geographical location, plastic pollution can have devastating effects on the environment.

Even though the terrestrial environment can act as a permanent sink of microplastics, research has been still lacking [7]. Some studies analysed microplastics in urban dust [7,25] or urban stormwater retention ponds [26] and scare are those studies developed in urban soils [27]. Therefore, this study aims to (1) quantify light-density MPPs in parks and recreational areas in Amsterdam, (2) analyse the size distribution of MPPs, (3) investigate whether there were differences in MPP concentration and size distribution between parks and recreational areas, neighbourhoods, or city districts in Amsterdam, (4) identify plastic types from hotspot samples, (5) evaluate the chemical and physical soil characteristics and their relationships with MPPs, and, finally, (6) predict MPP concentrations and chemical and physical soil characteristics for the entire city through a spatial distribution approach.

## 2. Materials and Methods

### 2.1. Study Area and Sample Collection

Soil samples were taken from 10 parks, 12 recreational places, and 1 city forest in the urban area of Amsterdam, the capital of The Netherlands (Figure 1). Those soils are known as peat soils [28]. Amsterdam is divided into seven city districts: Centrum, Nieuw-West, Noord, Oost, West, Zuid, and Zuidoost.

A total of 64,300 g of soil samples were collected within a regular grid (with a distance of 330 m between points), and 10 additional samples were randomly taken within 30 m of the pre-set locations. In total, 74 samples were taken from the top 30 cm soil layer using a cylindrical auger (diameter = 5 cm) made from steel to prevent contamination. The procedure of [29,30] was followed, since plastics accumulate in the upper 25/30 cm of arable soils, where 95% of microplastics are present. Each sample was transported within a paper bag (16 × 10 × 35 cm) to prevent contamination, and was then stored in a refrigerator at 5 °C until further laboratory analyses.

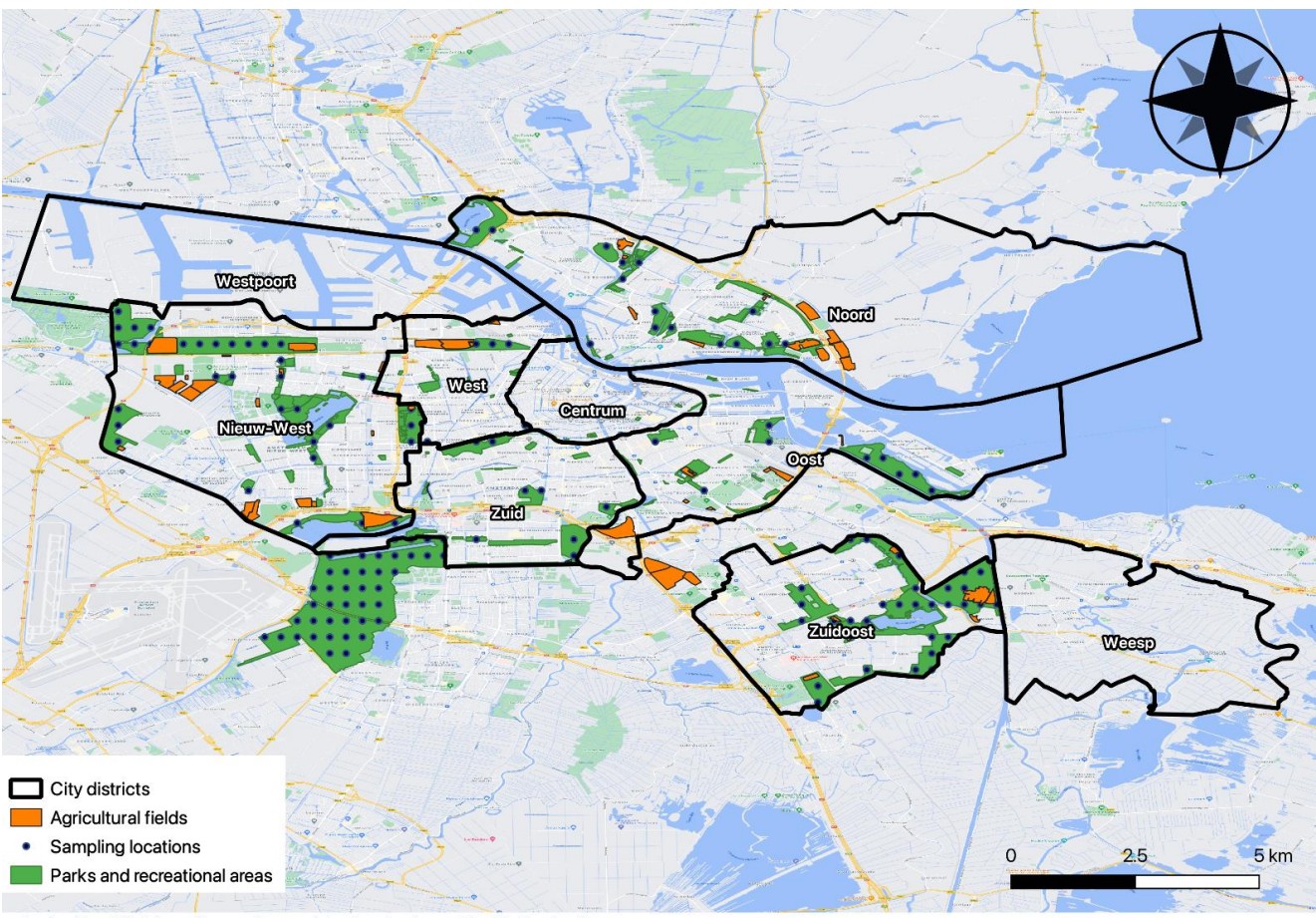

Agricultural fields and sampling points Amsterdam, The Netherlands

**Figure 1.** Sampling points in parks and recreational areas in Amsterdam, The Netherlands.

### 2.2. Microplastic Extraction

Microplastic extraction was based on the method proposed by Zhang [29], modified from [31] (in short, a flotation method based on density difference). First, all samples were air-dried in an oven at 40 °C for 24 h. Samples were gently pressed through a 2 mm sieve that was made from steel to prevent plastic contamination, weighted to $5.00 \pm 0.01$ g, and put into 50 mL glass tubes to prevent contamination.

A total of 40 mL of distilled water was added to the tubes and mixed in a Gerhardt Laboshake shake at 120 rpm for 30 to 40 min instead of 2 h, as proposed by Zhang [29] because the soil samples had less organic matter [32,33] and no hard aggregates. Second, samples were centrifuged at 3000 rpm for ten minutes. After centrifuging, the supernatant was filtered using Whatmann 91 filters with a pore size of <7 mm, which were made from paper to prevent contamination. Samples were topped up until they reached 50 mL again, and the process was repeated three times. Between iterations of filtering, all samples were covered with paper cloth to prevent contamination.

Afterwards, the samples were put into an ultrasound machine for two hours to break up any remaining aggregates. The samples were left overnight, covered with paper cloth to prevent contamination, and filtered. The filters were then dried in the oven at 60 °C for 3 h. After drying, the filters were folded shut to prevent contamination and were stored in a dark cupboard until the analyses.

### 2.3. Microplastic Constatation

To recognise microplastics, filter residue was brushed onto a glass plate using a finest red sable (da Vinci) paintbrush (number 5) with weasel hairs to prevent contamination. During constatation, the researcher wore mainly cotton clothes and a cotton coat. Two

pictures were taken using a ZEISS Stemi 508 microscope (1:8 zoom) before and after heating for 30 s at 140–150 °C. Microplastics were selected in Photoshop by using changes in shape, colour, or transparency after heating (Figure S1, according to Zhang [29]). Each selection was analysed for the plastic count and size using ImageJ.

### 2.4. Microplastic Identification

To identify the types of microplastics, a random subgroup from the sampling locations with the highest microplastic concentrations in the soils was assessed. It was decided to analyse a subgroup, since the identification method was highly time consuming. Seven sampling locations were analysed in triplicate; the locations were: 'Westerpark', 'Gerbrandypark', 'Flevopark', 'Nelson Mandelapark', and three from the 'Amsterdamse Bos', hereafter called Amsterdamse Bos 1, Amsterdamse Bos 2, and Amsterdamse Bos 3. Plastic was extracted by following the method explained in Section 2.2. After extraction, the 21 samples were identified using a Laser Direct Infrared Chemical Imaging System (8700 LDIR). After identification, a correlation matrix was made for all of the particles. When the correlation of the absorbance spectra of the plastic type in the sample and absorbance spectra of the plastic type from the LDIR library was above 0.85, the plastic type was considered a match.

### 2.5. Soil Characterization by Using Organic Carbon Content, Moisture Content, and pH

Soils were characterized by using organic matter content, moisture, and pH; the methods' descriptions are present in the Supplementary Materials.

### 2.6. Data Analysis

A Shapiro–Wilk test was performed to test for data normality, and it showed that the data were not normally distributed ($p < 0.05$). To determine if there were any statistical differences among parks and recreational areas, a Kruskal–Wallis test was performed. Afterwards, potential differences were identified using a post-hoc Mann–Whitney U test. To differentiate between the sizes of the particles, the particles were divided into four size groups: A: >1 mm, B: 0.50–1 mm, C: 0.25–0.50 mm, and D: <0.25 mm. Correlations between microplastic concentration, OCC, MC, and pH were determined using Spearman tests. All tests were performed in RStudio (Version 1.2.5) using a significance level of $p < 0.05$.

### 2.7. Geostatistical Analysis

Geostatistics were used to examine the spatial distribution structures of MPP concentration, organic carbon content, moisture content, and pH in soils. The main tool for the analyses was the semi-variance function; the semi-variogram (Figures S2−S9) value was calculated using Equation (1).

$$\gamma(h) = 1/2N(h) \sum_{mi=1}^{N(h)} [Z(xi) - Z(xi + h)]^2 \tag{1}$$

$\gamma(h)$, or the semi-variance value, is half of the expected squared difference between the value of *Z* at two locations separated by the distance interval *h*. The spatial structure was best fit by the spherical variogram model. The parameters used are:

- nugget variance: the random variation of the short distance.
- sill: the maximum value of the semi-variogram.
- range: the separation distance at probable spatial dependence.

The variogram was then fitted in R-studio (Version 1.2.5). The nugget-to-sill ratios and variograms can be found in the Supplementary Materials.

## 3. Results

### 3.1. Microplastic Particles in Soils by City District

Microplastic particles were found in 97% of the analysed samples. Most particles were found in 'Oost', and the lowest concentrations were found in 'Zuid Oost': 5996 ± 10,658

and $1198 \pm 2879$ MPP kg$^{-1}$, respectively (Tables S1 and S2). There were no significant differences between city districts ($p < 0.05$, Figure 2). Outliers were present in 'Amsterdamse Bos' and 'Zuid Oost'. The hotspot samples had around $78.7 \pm 12$ microplastics per gram of soil (these particles were identified in $380 \pm 95$ extracted and studied particles per gram of soil in the assessment with the LDIR).

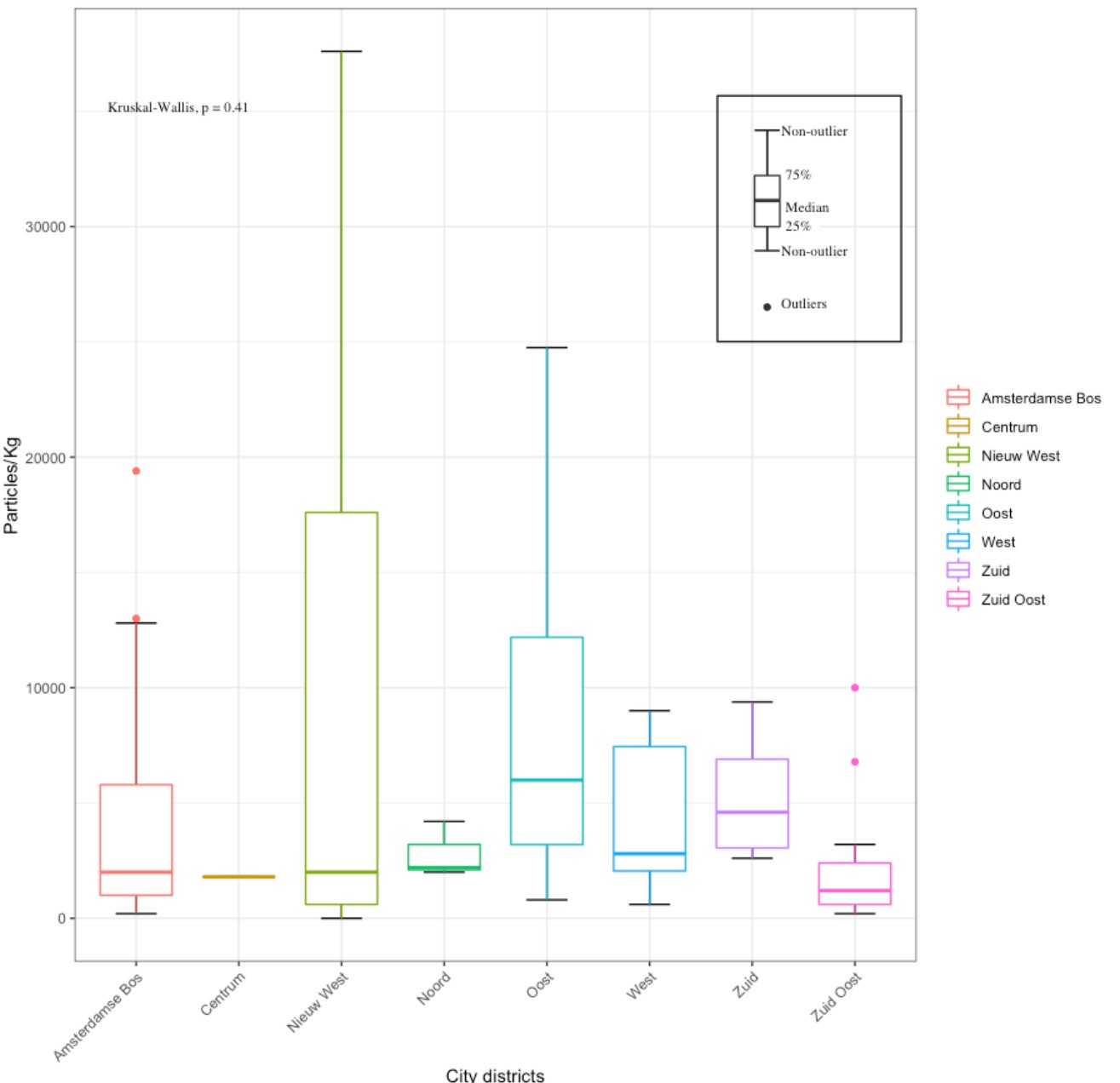

**Figure 2.** Visualization of microplastic concentrations (MPP Kg$^{-1}$) in soils from parks and recreational areas (n = 74) by city district in Amsterdam.

### 3.2. Microplastic Size Distribution by City District

The size distribution was analysed by city district (Figure 3). The highest significant median was found in 'Oost' (0.37 mm), and the lowest was found in 'Noord' (0.19 mm). The largest diameter was found in 'Oost' (0.40 mm). All city districts showed outliers. The Kruskal–Wallis test showed significant differences between city districts ($p < 0.05$).

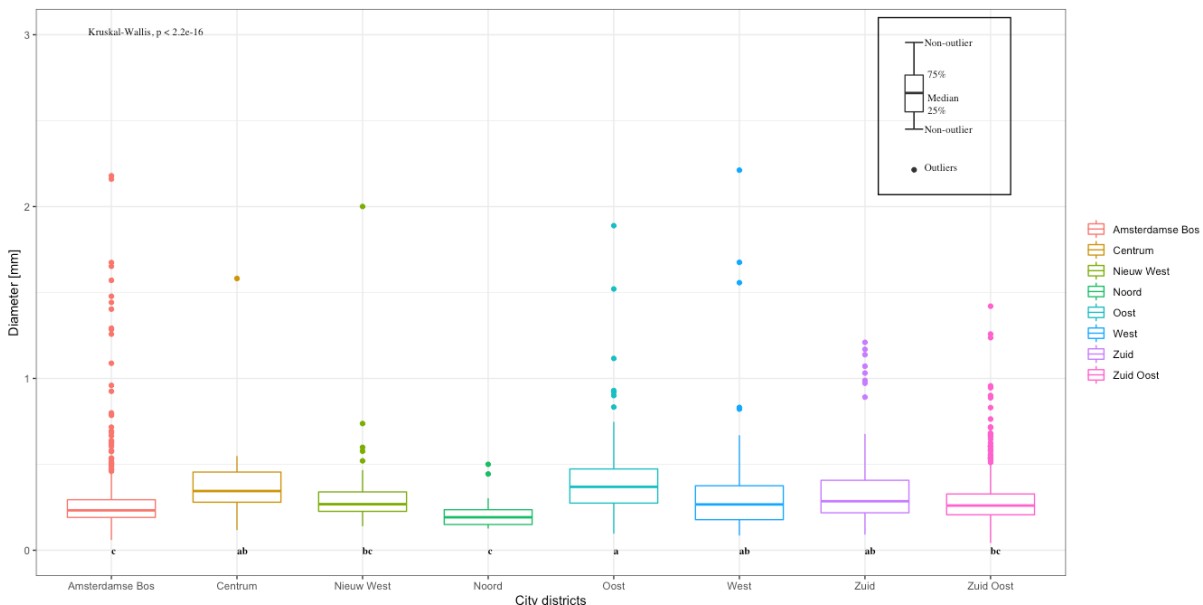

**Figure 3.** Size distribution (diameter in mm) of microplastic particles (n = 1748) in soils from parks and recreational areas (n = 74), which were analysed by city district in Amsterdam. Significant differences between city districts ($p < 0.05$) are indicated by lowercase letters (a > b > c).

### 3.3. Microplastic Types

Each sample contained low-density polyethylene (LDPE) (40.82%), natural polyamide (15.58%), pro-oxidant-additive-containing (PAC) plastics (36.36%), polypropylene (PP, 0.19%), polystyrene (PS, 1.30%), bioplastic (0.19%), polyamide (PA, 0.37%), polyurethane (PU, 0.56%), polyvinyl chloride (PVC, 0.19%), and unidentified plastics (3.71%). The plastic type with the highest concentration was LDPE, which was found in the Amsterdamse Bos 3 with 8800 particles/kg soil (Figure 4). The shapes and colours of the microplastics were not characterised.

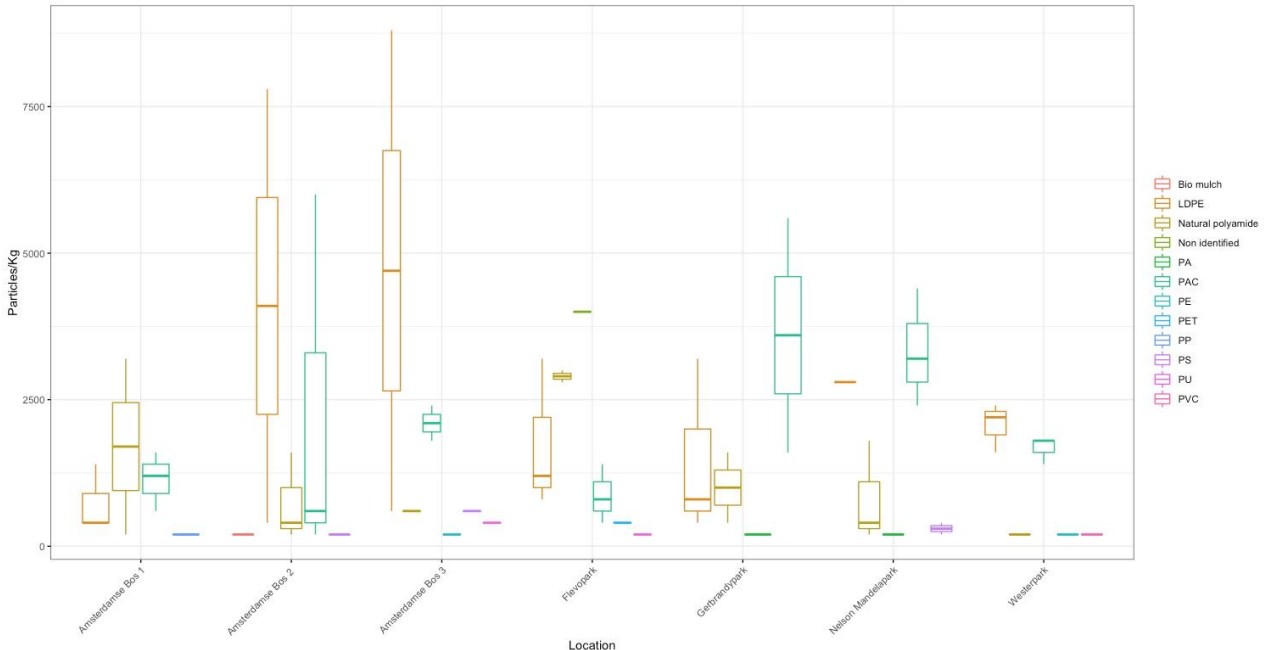

**Figure 4.** Plastic types in soils from parks and recreational areas that were analysed by city district in Amsterdam.

### 3.4. Soil Microplastics, Organic Carbon Content, Moisture Content, and pH Prediction in the Soils of Amsterdam

The microplastic concentration was predicted for the entire city using a spherical model in the semi-variogram (Supplementary data). Figure 5 shows that the soils, on average, contained 5000 MPP kg$^{-1}$ and were moderately spatially dependent, which could be concluded from the nugget-to-sill ratio (Supplementary data). A hotspot was observed in the western part of the city.

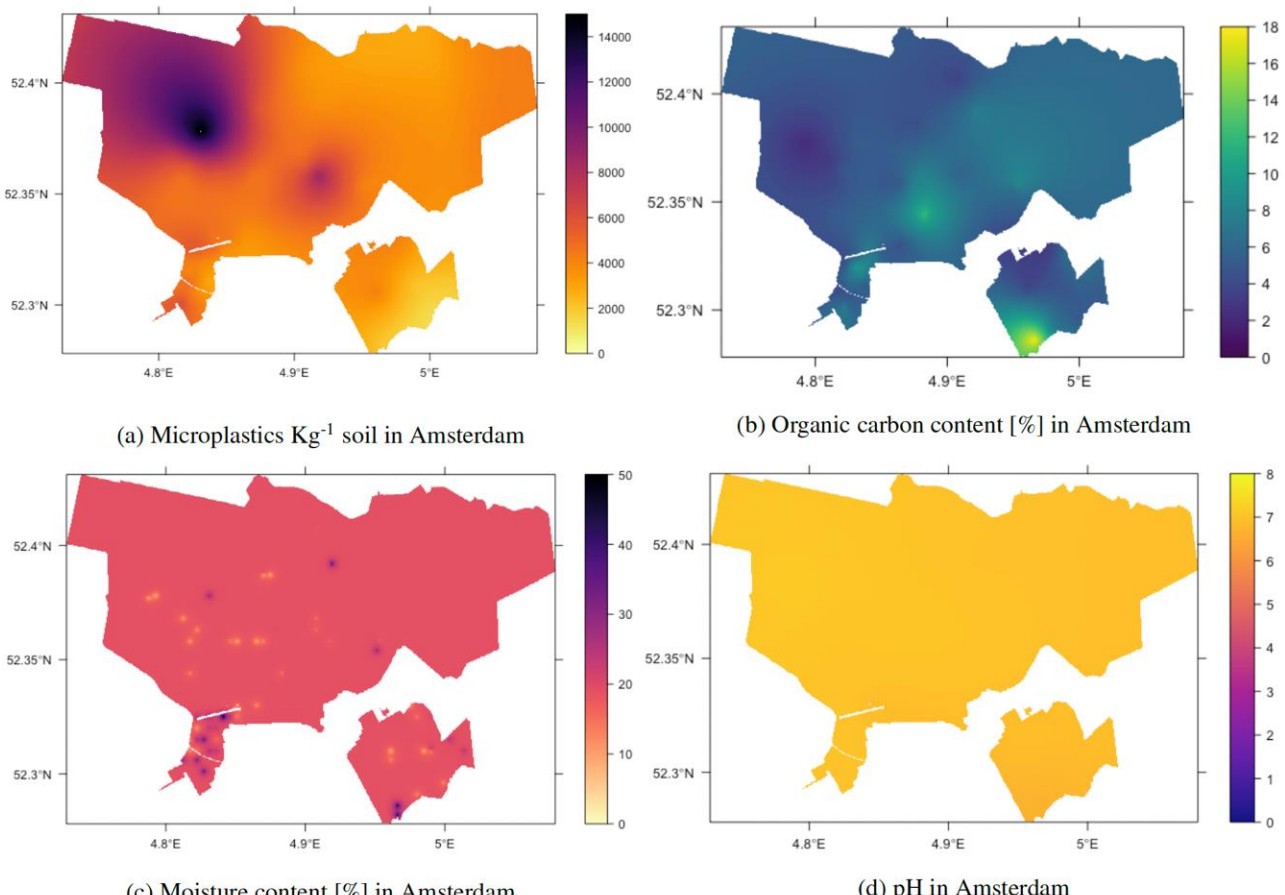

**Figure 5.** Prediction maps of (**a**) microplastic concentration (MPP/kg), (**b**) organic carbon content (%), (**c**) moisture content (%), and (**d**) pH in soils from parks and recreational areas of Amsterdam.

Organic carbon analyses showed that the highest median was found in the city district 'Oost' (8.09%), and the lowest was found in 'Noord' (2.35%). Outliers were found in 'Amsterdamse Bos', 'Oost', and 'Zuid Oost'. There were no significant differences between city districts. Consequently, the OC content was plotted for the entire city of Amsterdam (Figure 5b) using a spherical model. The nugget-to-sill ratio showed that the OC content was moderately to strongly spatially dependent. The mean OC content was 5.95%, and a higher OC content was roughly expected in the southern part of 'Zuid Oost', which is coloured yellow in the figure (Figure 5c). An insignificant ($p < 0.05$) negative correlation with microplastic concentration was found.

The city district with the highest moisture content was 'Noord' (20.18%), and the lowest was 'West' (4.71%). A significant difference ($p < 0.05$) between city districts was observed. Outliers were only observed in 'Amsterdamse Bos'. The mean moisture content was 18.89% for the entire city, and a higher moisture content was expected in the southern part of 'Zuid Oost' and the 'Amsterdamse Bos', which are coloured purple in the figure (Figure 5c). A non-significant ($p < 0.05$) negative correlation with microplastic concentration

was observed. The semi-variogram of the moisture content, which was plotted using a spherical model, showed that the data were moderately spatially dependent.

The highest median pH was found in the city district 'Centrum' (7.28). The lowest median pH was found in 'Noord' (6.89). Outliers were found in 'Amsterdamse Bos', 'Nieuw West', 'West', and 'Zuid Oost'. The pH showed no significant differences between city districts. The average pH was 6.96 and did not change much by city district (Figure 5d). The semi-variogram was plotted using a spherical model and showed that the data were moderately spatially dependent. The correlation was slightly positive, but insignificant.

## 4. Discussion

### 4.1. Microplastic Concentration in City Districts

It can be stated that MPPs are abundant in the soils of Amsterdam. In this study, 97% of the samples contained MPPs, with a mean of 4825.31 ± 6513.85 MPP/kg soil. The study by Huerta Lwanga [34] was the most similar in terms of methodology, where soil samples were analysed for microplastics in home gardens with a similar extraction method. In that research, 870 ± 1900 MPPs/kg were found in soil. These results are much lower than those calculated in this study. It is known that a denser population results in more plastic waste and litter [7]. Unfortunately, there are no exact data available on the amount of litter in Amsterdam. Nevertheless, it is estimated that the amount of litter in The Netherlands lies between 50 and 275 million kg per year. This includes data on litter that is cleaned by street cleaners [35]. Moreover, it is estimated that in Europe, annually, between 1270 and 2130 tonnes of microplastics/million inhabitants are unloaded in urban environments [6]. In Amsterdam, this would mean that each year, between 1043 and 1750 tonnes of microplastics are released into the environment. Other potential sources of microplastics in urban environments are tire abrasion, construction, atmospheric dispersion, and washing of synthetic clothing [7]. Generally, quantitative research on potential sources of microplastics in the environment is lacking. Studies are restricted by the complex sources, incomplete data on transport and fate in the general environment, and high geographical variability [36].

The highest microplastic concentration was found in 'Nieuw-West', and the lowest was found in 'Zuidoost'. The highest median, however, was found in 'Oost' (5996 ± 10,658 MPP/kg). Corresponding to the results in the parks and neighbourhoods, 'Oost' is crossed by a railway and a highway. This could explain the higher concentration of MPP/kg in the soil. Nevertheless, the differences between city districts were insignificant; it is possible that this was due to the low number of sampling locations per city district.

As mentioned before, it was proposed that a higher population density would result in a higher MPP/kg concentration. This does not correspond to our data, as 'Oost' does not have the highest number of residents.

### 4.2. Size Distribution of Microplastic Particles in Amsterdam

Most particles (on average, 47 ± 22%) had a diameter <0.25 mm. Plastic particles become so small due to several factors, such as embrittlement, photo-oxidation, abrasion, or fragmentation by UV light [37]. The distribution of the diameter of each MPP was categorized per city district. The largest median was discovered in 'Centrum' (0.48 mm). The environmental risk that this represents is related to the ingestion of MPPs by soil fauna. According to Huerta Lwanga [38], earthworms are able to bioturbate particles up to a size of 0.50–1 mm; nevertheless, it is mostly particles smaller than 0.25 mm that are found in their guts. When microplastics end up in earthworms' guts, they can result in weight loss, lower growth rates, and even mortality, thus decreasing soil quality [38].

Furthermore, when burrows created by earthworms become denser, organic matter concentrates in the burrow walls, which, in turn, can increase the inclination toward sorption of contaminants [38]. Moreover, the bioturbation of microplastics in earthworms can result in accumulation in the food chain, thus posing a threat for human health. Microplastics haven been found in several types of terrestrial animals [39], such as chickens [34].

Finally, it is suggested that terrestrial organisms and soil microbial communities may advance the degradation of plastics into microplastics or even nanoplastics. This can, in turn, stimulate the soil as a long-term sink of micro- and nanoplastics [14].

### 4.3. Microplastic Types

The types of microplastics were determined in seven sampling points from the areas with the highest concentrations of MPPs in soils. Each sample contained LDPE, natural polyamide, and PAC. LDPE is mostly related to packaging and is frequently used to produce film. Natural polyamide is a form of bioplastic, which is plastic derived from biomass and is most often produced by transforming cellulose into products that have thermoplastic properties, such as films and fibres [40]. Other natural polymers that can be used to produce natural polyamide are starch, polyhydroxyalkaboates (PHAs), or lignin [40]. PAC plastics are conventional fossil-based polymers containing additives to enhance polymer degradation when exposed to light and/or heat [41]. In this research, PAC plastic is LDPE with additives, which is often produced in the market as oxodegradable plastic bags or other single-use plastics. PE, PP, PVC, PET, PS, and PA are all considered thermoplastics, which melt when heated and harden when cooled [42]. PP and PE are mostly used in packaging, but are used in all sectors of primary plastic production worldwide [42]. PVC is mostly used in the construction sector, and is often used for pipes, hoses, and window and door frames [42], whereas PET is almost only used in packaging, mostly for bottles [43]. PUR is an example of a thermoset and is mostly used for foam products, such as insulation material. PS is used for food packaging, plastic cups, and cutlery [43]. It is noticeable that most of the recovered plastic types are used in either the packaging or construction industry. This coincides with findings by Geyer [42], where it was found that in 2017, the global primary plastic production was mostly made up by packaging (36%), followed by construction (16%). This could mean that most of the plastics recovered in the parks and recreational areas are related to littering or construction work.

### 4.4. Microplastics and Soil Organic Matter, pH, and Moisture

It was a challenge to try to understand if microplastics could be related to soil properties. The idea arose from trying to infer that soils that are rich in organic matter are those that have been under a continuous supply of fertilisers or composts (composts are also known to vehicles of microplastics [44]). Consequently, the following hypothesis can be addressed: In soils where compost was added intensively, will there also be high concentrations of microplastics? We cannot know, as in the present study, it was not possible to elucidate this from the soil data. It is necessary to study the land use and the constant application of composts, which was outside of the scope of this study. Further studies are needed to understand the spatial distribution of microplastics and its relation with soils that are rich in organic matter in the soils of parks and recreational areas in Amsterdam.

### 5. Conclusions

It can be concluded that microplastic particles are abundant in the soils of Amsterdam. Although there were no significant differences between city districts, understanding the differences could aid in policy making in order to minimize the negative environmental effects of microplastic pollution. A total of 53% of all particles were smaller than 0.25 mm, thus posing a threat to soil quality. Most plastic types that were recovered were related to the packaging and construction sector, indicating that the main sources of microplastics in the soil could be from littering and construction work. The relationship of microplastics with several soil properties is still insufficiently understood, but it is clear that the spatial distribution is determined by anthropogenic factors.

**Supplementary Materials:** The following are available online at https://www.mdpi.com/article/10.3390/environments9010005/s1, Figure S1: Pictures of plastic particles after and before heating. Figure S2: Variogram of MPP/Kg in Amsterdam, The Netherlands. Figure S3: Fitted variogram

of MPP/Kg in Amsterdam, The Netherlands. Figure S4: Variogram organic carbon content in Amsterdam, The Netherlands (before fitting). Figure S5: Fitted variogram organic carbon content in Amsterdam, The Netherlands. Figure S6: Variogram moisture content in Amsterdam, The Netherlands (before fitting). Figure S7: Fitted variogram of moisture content in Amsterdam, The Netherlands. Figure S8: Variogram of pH in Amsterdam, The Netherlands (before fitting). Figure S9: Fitted variogram of pH in Amsterdam, The Netherlands. Table S1: Statistical data of microplastic particles per kg per city district in Amsterdam. No significant differences between city districts were observed after a Kruskal-Wallis test. Table S2: Microplastic particles [MPP/Kg] per park or recreational area in Amsterdam, The Netherlands.

**Author Contributions:** Conceptualization, E.H.-L. and Q.M.C.; methodology, Q.M.C., K.M. and M.G.; validation, all authors participated; formal analysis, Q.M.C. and M.G.; investigation, Q.M.C., M.G. and E.H.-L.; resources, E.H.-L. and V.G.; data curation, Q.M.C.; writing—original draft preparation, Q.M.C. and E.H.-L.; writing—review and editing, Q.M.C., E.H.-L. and V.G.; visualization, Q.M.C. and E.H.-L.; supervision, E.H.-L.; project administration, E.H.-L.; funding acquisition, V.G. All authors have read and agreed to the published version of the manuscript.

**Funding:** This research received no external funding.

**Acknowledgments:** We would like to acknowledge Ir. Gerard Heuvelink and Cynthia van Leeuwen for their advice on geostatistics.

**Conflicts of Interest:** The authors declare no conflict of interest.

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
