# Peer review of "Parks and Recreational Areas as Sinks of Plastic Debris in Urban Sites: The Case of Light-Density Microplastics in the City of Amsterdam, The Netherlands"

_environments, doi:10.3390/environments9010005_

Round 1

Reviewer 1 Report

  • The article is interesting, since it reports the level of microplastic contamination in urban soils, a surely understudied field of research which deserves further investigation. A minor revision of the English language is required, since some passages are quite unclear. A few issues were also highlighted during review: 
    - The main issue is that no information about contamination control is reported in this manuscript. External contamination is a major issue in microplastic research, and this has been the focus already of countless publications. In this paper, no information is reported about contamination control measures adopted during sampling and or sample processing and analysis. No procedural blanks were carried out and the level of external contamination are not reported. Please address this serious issue. 
    - Secondly, I believe that the total number of microplastic particles extracted from the samples needs to be reported, as well as the total number of particles that were chemically characterize, so that an estimate of the representativeness of this subset can be drawn. Please report total number of particles extracted, counted and analyzed. 
    - Lines 177-182: PAC is not reported here. What is PAC? Please explain this acronym and provide more information, besides that already reported at lines 271-272. What is this? 
    - Also, it is not entirely clear to me what is a "natural polyamide"? Is this cellulose? Can you explain better? Polyamide is different from Nylon? Please provide more details. 
    - Lastly, information about the shape, type and colors of the extracted particles need to be reported. How many textile fibers? How many pellets, etc? Please follow more rigorous reporting guidelines, such as those reported by: 
  • https://doi.org/10.1364/AS.74.001066
  • https://doi.org/10.1177%2F0003702820938993
  • https://doi.org/10.1016/j.scitotenv.2020.141426
  • https://doi.org/10.1177/0003702820945713

Author Response

Answers to reviewers.

Dear reviewers, we appreciate your suggestions and comments, please find the following answers.

Reviewer 1.

The article is interesting since it reports the level of microplastic contamination in urban soils, a surely understudied field of research that deserves further investigation. A minor revision of the English language is required, since some passages are quite unclear. A few issues were also highlighted during review: 
- The main issue is that no information about contamination control is reported in this manuscript. External contamination is a major issue in microplastic research, and this has been the focus already of countless publications. In this paper, no information is reported about contamination control measures adopted during sampling and or sample processing and analysis. No procedural blanks were carried out and the level of external contamination are not reported. Please address this serious issue. 

Dear reviewer thank you for your words, now it is added in the document the measures taken against external contamination.

- Secondly, I believe that the total number of microplastic particles extracted from the samples needs to be reported, as well as the total number of particles that were chemically characterize, so that an estimate of the representativeness of this subset can be drawn. Please report total number of particles extracted, counted and analyzed. 

Dear reviewer we appreciate your interest and question, the scope of  the Zhang et al. method is to identify the plastic particles, by optic microscope, and it was not possible to count all the particles present in the sample. Once we did the identification of the type of plastic by the LDIR in the hotspots samples, then it was possible to count the total number of particles extracted per sample, being an average of 380±95 particles extracted per gram of soil, and from them an average of 78.7±12 particles were microplastics per gram of soil, being around the 20% of the extracted particles microplastics, this information is added in the document.

- Lines 177-182: PAC is not reported here. What is PAC? Please explain this acronym and provide more information, besides that already reported at lines 271-272. What is this? 

Done, the full name was added.

- Also, it is not entirely clear to me what is a "natural polyamide"? Is this cellulose? Can you explain better? Polyamide is different from Nylon? Please provide more details.

The information is added

- Lastly, information about the shape, type and colors of the extracted particles need to be reported. How many textile fibers? How many pellets, etc? Please follow more rigorous reporting guidelines, such as those reported by:

Dear reviewer the objective of the study was to inform over the light density microplastics found in soils, the objective was not to characterize all the microplastics particles, so we thank you very much for your comment, and that is also comment in the document.

Reviewer 2 Report

Manuscript titled: "Parks and recreational areas as sinks of plastic debris in urban sites. The case of light density microplastics in the city of Amsterdam, the Netherlands. " has been reviewed critically. The revised manuscript shows very interesting research on quantification and determination of the type and abundance of light density microplastics in the soil in parks and recreational areas of Amsterdam, Netherlands. Following are the major points to consider before considering for publication.

1. Although the paper is very interesting and deals with environmentally essential aspects related to the placement and origin of microplastics in an urban environment, it still requires additional work before publication.

2. It would be worthwhile to indicate in Figure 1 the location of the places from which the samples were analyzed. In addition, the quality of work is significantly reduced by poor quality Figures 2, 3, and 5. They must be replaced with better quality once. Moreover, Figure 5b in the text is incorrectly signed (line 201).

3. There is a massive mess with citations; their numbering does not agree with the list at the end of the paper (e.g., reference 31). The work is undeveloped; there are numerous double spaces in the text. I recommend revising the writing and language throughout the document: verbs are missing in some phrases, poorly written sentences, etc. As far as I find research designed appropriate and clearly described, however, the quality of its presentation requires improvement.

I suggest reconsidering the paper for publication after revision.

Author Response

Thank you for your kind words.

It would be worthwhile to indicate in Figure 1 the location of the places from which the samples were analyzed.

The sampling points are indicated in figure 1, now the figure is presented with better quality, so it is possible to appreciate the sampling locations. 

There is a massive mess with citations; their numbering does not agree with the list at the end of the paper (e.g., reference 31).

All citations were now checked and corrected. Apologies for this inconvenient. 

The work is undeveloped; there are numerous double spaces in the text. I recommend revising the writing and language throughout the document: verbs are missing in some phrases, poorly written sentences, etc.

All the document is checked and revised.  

In addition, the quality of work is significantly reduced by poor quality Figures 2, 3, and 5. They must be replaced with better quality once. Moreover, Figure 5b in the text is incorrectly signed (line 201).

New pictures were provided.
